# Effective Adsorption of Diesel Oil by Crab-Shell-Derived Biochar Nanomaterials

**DOI:** 10.3390/ma12020236

**Published:** 2019-01-11

**Authors:** Lu Cai, Yan Zhang, Yarui Zhou, Xiaodie Zhang, Lili Ji, Wendong Song, Hailong Zhang, Jianshe Liu

**Affiliations:** 1College of Environmental and Science Technology, Donghua University, Shanghai 201620, China; lucai89@126.com; 2College of Food and Medical, Zhejiang Ocean University, Zhoushan 316022, China; zhangyanyyqx@163.com; 3College of Port and Transportation Engineering, Zhejiang Ocean University, Zhoushan 316022, China; zyr0612Z@163.com (Y.Z.); thanxfoevercoming@163.com (X.Z.); 4Institute of Innovation & Application, Zhejiang Ocean University, Zhoushan 316022, China; jll-gb@163.com; 5College of Petrochemical and Energy Engineering, Zhejiang Ocean University, Zhoushan 316022, China; swd60@163.com

**Keywords:** crab shell, activated biochar, adsorption performance, oily wastewater

## Abstract

This study, for the first time, rendered crab shell activated biochar modified by potassium hydroxide (KOH) impregnation (CSAB), revealing a new potential application in the removal of diesel oil from oily wastewater. The structural characteristics of crab shell biochar (CSB) and CSAB were investigated by SEM, and the crystal structure and optical properties of as-prepared samples were analyzed using XRD and FTIR. Results showed that CSAB had stratified surface structure morphology, abundant functional groups, and that its high specific surface area could reach up to 2441 m^2^/g, which was about eight times larger than that of untreated CSB (307 m^2^/g). An adsorption isotherm study indicated that the actual adsorption process both of CSAB and CSB were found to fit better with the Freundlich equation. Moreover, chemical interaction controlled the adsorption kinetics efficiency while the adsorption equilibrium capacity was 93.9 mg/g. Due to its highly developed pore structure, unique surface characteristics, and effective adsorption performance, this low-cost activated carbon had the potential to serve as an efficient adsorbent for water pollution purification.

## 1. Introduction

Over recent years, large amounts of diesel oil hydrocarbon have entered the environment as a result of human industrial activities such as oil extraction, refining, storage, transportation, and production of the petrochemical industry [1,2]. The residue of oil pollutants in natural water has aroused the wide attention of researchers [3]. Hydrocarbon contamination may be one of the serious consequences resulting from the discharge of such oily wastewater, which induces considerable harm to the survival of marine life, the atmospheric system, and human health. Particularly, the extinction of marine life and the frequent occurrence of natural disasters have made us aware of the urgency of marine pollution control and that we must take corresponding countermeasures as soon as possible [4,5,6,7].

Nowadays, various treatment methods for oily wastewater can be divided into five categories: dissolved air floatation, mechanical separation, chemical, physicochemical, and biological methods [8,9]. These techniques are found to be possibly effective for large-scale or certain concentrations of oil pollution. However, there are still some restrictive factors for the utilization of these methods, like secondary pollution to the environment, high cost, and complicated operation processes involved [10]. The adsorption technique—as an economical and excellent approach for the removal of organic pollutants—has drawn much research awareness in many environmental fields [11]. The application of the adsorption process to treat oily wastewater effluents with activated carbon is a promising alternative treatment method that is scarcely explored [12]. In general, activated carbons are the most widely used adsorbents in both granular and powdered forms due to their excellent adsorption capability for organic matter, which is usually related to their porosity, specific surface area, and pore volume [13,14,15]. For example, the adsorption performance of methylene blue onto activated carbon prepared from date pits has been explored [16]. Carbon residue (or char) produced as a by-product from woody biomass gasification has been suggested to have a high adsorption capacity of Rhodamine B (RhB) dye, mainly due to the wide specific surface area and abundant functional groups on the activated carbon surface [17]. Not only do carbon materials absorb organic dyes, but they can also be utilized in the treatment of oil pollution. Facile preparation of carbon foams has been used to remove emulsified oil from aqueous solutions and the kinetics and equilibrium of the adsorption process have also been studied and analyzed [18]. Okiel et al. investigated the ability of adsorbents to adsorb oil and different factors affecting adsorption properties [19]. Prajapati et al. studied the effects of adsorptive desulfurization of diesel oil on nickel nanoparticle-doped activated carbon beads with or without carbon nanofibers [20]. All of the above research demonstrates the great potential of carbon materials as adsorbents for removal of organic pollutants.

Crab shell attracted considerable attention for the production of porous carbon adsorption material. Rae et al. investigated the fast and excellent adsorption performance of easily processed adsorbents derived from crab shells for the removal of Hg from acidic solutions [21]. Lu et al. carried out an experiment on the effective removal of zinc ions from aqueous media onto milled crab carapaces [22]. The sorption of nickel on crab shells was accomplished by ion exchange and a chelating mechanism [23]. Jeon determined that waste crab shell has a high potential to remove arsenate ion up to about 35.92 mg/g [24]. Dai et al. explored the physicochemical properties and removal potential of calcium-rich biochars (CRB) prepared through pyrolysis of crab shells at various temperatures [25]. As far as we know, there is no report on the research of crab shell biomass for oil adsorption. To produce a valuable adsorbent with low cost, excellent surface structure, and high absorption performance of oil pollutions, we chose crab shells as a precursor for the preparation of activated carbon and modified by potassium hydroxide (KOH) impregnation in previous studies [26,27,28].

In this paper, the possibility of modified crab shell biochar prepared at suitable pyrolysis temperatures as a novel adsorbent in the removal of oily wastewater was investigated. Specifically, the physiochemical characteristics of samples were analyzed by XRD, SEM, BET, and FTIR. The adsorption thermodynamics and adsorption kinetics were studied as the assessment indexes of the potential for removal of the organic pollutants from waste water.

## 2. Materials and Methods

### 2.1. Materials

The crab (*Portunus trituberculatus*) shell was collected at Zhoushan, Zhejiang, China. HCl, KOH, and C_5_H_12_ were purchased from Sinopharm Chemical Reagent Co., Ltd., Shanghai, China. Diesel oil 0# was purchased at China Petrochemical Corporation, Zhoushan, Zhejiang, China. All chemicals were analytical grade and used as received without further purification.

### 2.2. Preparation of Activated Carbon

The crab shell was first washed to remove dirt from its surface and treated with 2 mol/L HCl for 6 h, then washed with distilled water until neutral and dried at 80 °C. The dried crab shell loaded in corundum crucible was placed in a tube furnace (Shanghai Jvjing Precision Instrument Manufacturing Co., Ltd., Shanghai, China). It was heated at a rate of 10 °C/min from room temperature to 700 °C under purified nitrogen (99.999%) flow of 100 mL/min. The carbonization step was held for 2 h at 700 °C [29]. Subsequently, the sample was transferred to a beaker containing 2 mol/L HCl, stirred for 1 h, then washed with deionized water and dried at 80 °C. The obtained carbon sample was named as CSB. Then, the CSB was mixed with KOH at an impregnation ratio of 3:1 [KOH (g):carbon (g)]. The mixture was pyrolyzed in a tube furnace under nitrogen flow of 100 mL/min to a final temperature of 800 °C for 1 h. The activated product was then cooled to room temperature under nitrogen flow and washed with deionized water. After being dried and ground into powder, the activated biochar was prepared and labeled as CSAB (crab shell activated biochar modified by potassium hydroxide).

### 2.3. Characterization

The phase structures of the as-prepared samples were investigated by XRD analysis at room temperature on an XRD powder diffraction instrument (D8 ADVANCE Da Vinci, BRUKER AXS GMBH, Karlsruhe, Germany) with the range of 2θ from 20° to 80°. Specific surface area measurements were performed on a Micromeritics ASAP 2010 instrument (Micromeritics Instrument Ltd., Atlanta, GA, USA) and analyzed by the BET method. The morphology and the microstructure of the activated biochar were studied by field emission scanning electron microscopy (SEM, Hitachi-4800, Hitachi Co., Ltd., Tokyo, Japan). FTIR spectra were obtained using a Perkin Elmer Fourier transform infrared (Nicoletteis 50, ThermoFourier, Waltham, MA, USA) spectrometer with KBr as a diluting agent and operated in the frequency range of 4000–1000 cm^−1^.

### 2.4. Adsorption Experiments

Adsorption equilibrium experiments were conducted in batch mode in a series of 250 mL Erlenmeyer flasks containing 200 mL of distilled water with different initial concentrations in the range of 100–500 mg/L of diesel oil, 100 mg/L for the adsorption kinetics studies, and 100–500 mg/L for the isotherm studies. The Erlenmeyer flasks were sealed with plastic wrap and placed on a thermostatic oscillator that was oscillated at a constant temperature at a 150 r/min frequency for 30 min to obtain oily wastewater. Moreover, the effects of sorbent dosage, initial pH, contact time, and temperature on the removal efficiency of diesel oil were also investigated to value the adsorption property of the samples. The adsorbent dosage used in this study ranged from 0.1–0.5 g. The contact time of the adsorption of the experiments was from 10 to 360 min. According to this experiment, 0.2 g of crab shell activated biochar was added to the oily wastewater, and each sample was kept in an isothermal shaker of 150 r/min at different temperatures (0–50 °C) for 240 min to reach adsorption equilibrium. The initial pH of the reaction solution was adjusted to 2.0, 4.0, 6.0, 7.0, 8.0, and 10.0 by adding either diluted HCl or NaOH (0.1 mol/L). A similar procedure was followed for another set of Erlenmeyer flasks containing the same diesel oil concentration without activated biochar to be used as a blank. Then, the supernatant was extracted with petroleum ether and the absorbance was measured using a UV-vis spectrophotometer at a 256 nm wavelength. According to the absorbance calculate, the adsorption capacity of crab shell activated biochar by the following Equations (1) and (2), respectively:
(1)$$q=\frac{(C_{0}-C)\cdot V}{m},$$
(2)$$\%\mathrm{Removal}=\frac{100(C_{0}-C)}{C_{0}},$$
where *q* was the amount of diesel oil taken up by the adsorbents (mg/g), *C*_0_ and *C* (mg/L) were the liquid-phase concentrations of diesel oil at initial and equilibrium, respectively, *V* (L) was the volume of the solution, and *m* (g) was the mass of the dry adsorbent used.

The procedures of kinetic experiments were basically identical to those of the equilibrium test. The aqueous samples were taken at preset time intervals and the concentrations of diesel oil were similarly measured.

## 3. Results and Discussion

### 3.1. Structure Characterization of Crab Shell Biochar

The pore structure of the char was estimated using the nitrogen adsorption isotherm shown in Figure 1a. The nitrogen adsorption isotherm of CSB corresponded to type V adsorption isotherm with a hysteresis loop at pressure (P/P_0_ > 0.4). The adsorption capacity increased slowly at a low relative pressure (pressure < 0.05), indicating the existence of mesoporous [30,31]. Moreover, the CSAB belonged to the type IV adsorption isotherm [32,33], indicating that there may have been a certain number of micropores and abundant mesoporous in the modified crab shell biochar. A comparison of the nitrogen isotherms of CSAB with CSB revealed that activation using KOH produced a higher nitrogen adsorption capacity. Figure 1b shows the pore size distribution curve of crab shell biochar obtained by the Barrett-Joyner-Halenda (BJH) method [34]. The pore size distribution (PSDS) of CSB and CSAB also reflected the above results, which were mainly in the range of 1–20 nm.

Table 1 shows the calculated BET results of CSB and CSAB. It can be seen that the KOH activation increased surface area and the amount of pore volume with reduced pore sizes. The specific surface area of CSAB increased from 307 m^2^/g to 2441 m^2^/g, nearly 20 times larger than that of unmodified biochar. The total pore volumes also increased from 0.324 to 1.682 m^3^/g, indicating that the formation of uniform pores was well developed.

As seen from Figure 2, the microstructure of crab shell power showed a fibrous structure, and its surface was loose with uneven arrangement (Figure 2a,b). The layered and porous structure of CSB was obvious after being calcined. The pore size was uniform and loose (Figure 2c,d). After activation by KOH, the surface of biochar showed an exfoliated and stratified structure, and there were many depressions, dents, and cracks, which promoted the increase in the surface area (Figure 2e,f). This may have been due to the etching of KOH on the surface of biochar [35,36,37].

Figure 3 illustrates the FTIR spectra of CSB and CSAB. It can be seen that there was no significant change between crab shell biochars before and after modification, and the position of the main absorption peak had not changed. The bonds around 3450 cm^−1^ corresponded to the O–H stretching vibration of carboxylic groups, hydroxyl groups, phenolic groups, and water. The adsorption peak at 1380 cm^−1^ and 1620 cm^−1^ corresponded to the bending vibration of –OH and the stretching vibrations of C=O, respectively. The peaks mentioned above were found in two samples, though the band intensity of CSAB was weaker. Weak bands that appeared in CSAB at 2910 cm^−1^ were assigned to the C–H vibration in the methyl group. However, such a peak was not observed for the CSB sample, indicating that the organic matters decomposed completely after the KOH modification. Above all, the modification of crab shell biochar did not significantly change the functional groups on the surface, but the adsorption capacity of modified biochar on diesel oil was improved to some extent, which may have been caused by the increase in specific surface areas.

The crystalline structure and the purity of the as-prepared photo-catalysts were characterized by powder XRD. Figure 4 presents the XRD patterns of CSB and CSAB. From the infrared spectra of crab shell biochar before and after activation, the diffraction peaks located at about 23.0°, 29.3°, 35.9°, 39.3°, 43.1°, and 47.5° could be perfectly indexed to the (012), (104), (110), (113), (202), and (018) crystal planes of calcite according to the JCPDS database No. 81-2027. As for the pattern of CSAB, two main peaks were located at 2*θ* = 25° and 45° that could have been indexed to amorphous carbon.

### 3.2. Effect of Adsorbent Dosage

The effect of adsorbent dosage for the adsorption of diesel oil was studied by using CSB and CSAB adsorbents ranging from 0.1 to 0.5 g and fixing the pH value, initial diesel oil concentration, and adsorbent temperature at 7, 100 mg/L, and 30 °C. It was observed that when the mass of each adsorbent was 0.2 g (S/L ratio = 1:1000), the adsorbent reached its highest amount of diesel oil adsorbed by the adsorbent (Figure 5a,b). As adsorbent quantities were higher than this value, the removal rate of diesel oil began to decline. With the increase in adsorbent quality, the increase of the diesel removal percentage could have been attributed to the increase in the adsorbent surface area and the number of adsorption sites available for adsorption, as reported in many studies [38,39]. The decrease in the removal rate of diesel oil was possibly due to many factors, such as availability of the solute, interference between binding sites, electrostatic interactions, and reduced mixing due to a high adsorbent concentration in the solution [40].

### 3.3. Effect of Initial pH

One of the most important factors in adsorption studies is the effect of medium pH. Different species exhibit different pH ranges depending on the use of adsorbent. The effect of pH on the adsorption of diesel oil at equilibrium by biochar is shown in Figure 6. When pH value was increased from 2 to 10 for CSB and CSAB, the adsorbent dosage, the adsorbent temperature, and the contact time were 0.2 g, 30 °C, and 240 min, respectively. Under acidic conditions, the adsorption capacity was found to increase with increasing pH. When the pH was 7, the adsorption capacity reached the maximum, and the adsorption capacities of CSB and CSAB reached 82.3 and 62.4 mg/g, respectively. When the pH value continued to rise and the solution system became alkaline, the adsorption capacity declined. These results indicated that alkaline and acidic environments were not suitable for the adsorption process of biochar, and that the optimum pH for adsorbing diesel oil by crab shell biochar was 7. This variation was explained by the change in the amount of protons presented in the solution and the stability of oil emulsion. At a lower pH, a huge amount of protons were available, and they may have saturated the adsorbent sites, thus increasing the cationic properties of the adsorbent surface [41]. A strong acidity caused oil coalescence and subsequently increased the size of oil droplets [42]. As the initial pH > 7, the degree of solubility of diesel oil increased due to the existence of acidic species, which made it difficult for diesel oil to adsorb on biochar. With the increase in the initial pH, the excess of hydroxyl ions and diesel oil competed for the active adsorption sites in the solution [43].

### 3.4. Effect of Adsorbent Temperature

Figure 7 illustrates the adsorption capacity of diesel oil on the as-prepared biochar versus temperature. The adsorbent temperature ranged from 0 to 50 °C, and the adsorbent dosage, pH value, and contact time were 0.2 g, 7, and 240 min, respectively. It was found that the adsorption capacities of CSB and CSAB reached 64.4 and 90.4 mg/g, respectively, with increased temperatures from 0 to 30 °C. As the temperature continued to increase, the adsorption capacity decreased. Similar trends were reported by Chandra et al. for the adsorption of methylene blue on activated carbon prepared from durian shell [44]. It was explained that as temperature increased, the physical bonding between the organic compounds and the active sites of the adsorbent weakened [45]. The solubility of the diesel oil also increased, which caused the interaction forces between the solute and the solvent to become stronger than between the solute and the adsorbent. Therefore, the solute was more difficult to adsorb.

### 3.5. Effect of Contact Time

Figure 8 shows the adsorption capacity of biochar versus contact time. Experiments were performed at an adsorbent dosage of 0.2 g, a pH value of 7, and a temperature of 30 °C with different contact times that ranged from 10 to 360 min. As can be seen from Figure 8, the amount of diesel oil adsorbed on biochar increased with time. At a certain time point, it reached a constant value beyond which the diesel oil could not be further removed from the solution. At this point, the amount of diesel oil from activated carbon and the amount of diesel oil adsorbed on biochar were in a state of dynamic equilibrium. In this study, the contact time for CSB and CSAB to reach equilibrium was about 240 min.

### 3.6. Adsorption Kinetics

In order to investigate the mechanism of adsorption and to determine the rate-controlling step, the following kinetic rate equations were used to test the experimental data.

Pseudo-first order dynamic equation:
(3)$$\ln(q_{e}-q_{t})=\ln q_{e}-k_{1}t,$$

Pseudo-second order dynamic equation:
(4)$$\frac{t}{q_{t}}=\frac{1}{k_{2}{q_{e}}^{2}}+\frac{t}{q_{e}},$$
where *q_e_* (mg/g) and *q_t_* (mg/g) were the amount of adsorbed diesel oil at equilibrium and at time *t*, respectively, *k*_1_ was a rate constant of pseudo-first-order adsorption, and *k*_2_ was a rate constant of pseudo-second-order adsorption. The slop and intercept of the plot of ln(*q_e_* − *q_t_*) versus *t* were used to determine the first-order rate constant *k*_1_. The slop and intercept of the plot of *t*/*q_t_* versus t were used to calculate the second-order rate constant *k*_2_.

Two main adsorption kinetics models were employed to value the adsorption capacities of the samples. As shown in Figure 9 and Figure 10, the true adsorption process of CSAB was better described by the pseudo-second-order model (*R*^2^ > 0.99) than the pseudo- first-order model (*R*^2^ = 0.979), as was the adsorption process of CSB. Furthermore, the calculated *q_e_* values (CSB: 73.1 mg/g, CSAB: 93.9 mg/g) were much closer to the experimental *q_e_* values (CSB: 69.4 mg/g, CSAB: 89.4 mg/g) as listed in Table 2, indicating that the pseudo-second-order model fit for the actual adsorption process better. It can be also inferred that the adsorption process was controlled by chemisorption, which involved valence forces through the sharing or exchange of electrons between the adsorbent and adsorbate [46,47]. Studies have shown that oil adsorption is closely related to functional group properties of the sorbent, including O–H, C=O, and C–O [48,49,50]. According to the FTIR data, the as-prepared sample surfaces were rich in these functional groups. Thus, such surfaces rendered a good oil adsorption performance. Sokker et al. [51] investigated the adsorption of crude oil using hydrogel of chitosan-based polyacrylamide and their results also obeyed the pseudo-second-order kinetic model rather than the pseudo-first-order kinetic model. Their experimental results were similar to the present work.

Between CSB and CSAB, CSAB had better oil adsorption performance. Ngarmkam et al. studied native palm shell and its activated carbons, and their oil sorption capacity reached 30–90 mg/g [52], which is similar to this study. The adsorption capacity of wood and kenaf ranges from 4000 to 8000 mg/g, while that of cotton, kapok, and milkweed ranges 30,000–40,000 mg/g [53]. However, crab shell biochar has the advantage of being a waste resource that is available at low costs, as it is a by-product of aquatic products processing.

### 3.7. Adsorption Isotherms

The adsorption isotherm is important to describing how solutes interact with adsorbents, and is critical in optimizing the use of adsorbents. The Langmuir isotherm assumes a monolayer adsorption onto a surface containing a finite number of adsorption sites of uniform strategies and of adsorption with no transmigration of adsorbate in the plane of the surface.

The linear form of the Langmuir isotherm can be represented by the following equation:
(5)$$\frac{c_{e}}{q_{e}}=\frac{1}{k_{L}q_{m}}+\frac{c_{e}}{q_{m}},$$
where *c_e_* is the equilibrium concentration of the adsorbate (mg/L), and *q_e_* is the amount of adsorbate adsorbed per unit mass of adsorbent (mg/g). *K_L_* and *q_m_* are the Langmuir isotherm constants related to adsorption capacity and rate of adsorption, respectively.

The Freundlich isotherm can be applied to non-ideal adsorption on heterogeneous surfaces as well as multilayer sorption and is expressed as follows:
(6)$$\mathrm{lg}q_{e}=\mathrm{lg}k_{L}+\frac{1}{n}\mathrm{lg}c_{e},$$
where *c_e_* is the equilibrium concentration of the adsorbate (mg/L), *q*_e_ is the amount of adsorbate adsorbed per unit mass of adsorbent (mg/g), and *K_F_* and *n* are the Freundlich constants related to the adsorption capacity and adsorption intensity, respectively.

Figure 11 shows the equilibrium adsorption isotherm of diesel oil on as-prepared crab shell biochars and thermodynamic parameters are listed in Table 3. It can be seen from the diagram that the adsorption isotherm of CSAB was closer to a straight line when compared with CSB. As the concentration of the diesel solution increased, the adsorption capacity of CSAB to the diesel solution increased continuously. When the diesel oil concentration was 500 mg/L, the maximum adsorption capacity of CASB reached 403.3 mg/g. However, the adsorption capacity of CSB increased slowly when the diesel solution concentration reached 400 mg/L. It may have been that diesel oil particles covered the surface of biochar, resulting in the decrease in the adsorption rate [54]. It can be seen clearly in Figure 11 that the adsorption capacity of modified crab shell biochar was significantly stronger than that of unmodified crab shell biochar.

Two isothermal adsorption models were employed to analyze the nature of the adsorption process. As shown in Figure 12 and Figure 13, the Freundlich model yielded the best fit, with *R*^2^ values equal to or higher than 0.99 when compared to the Langmuir model. Therefore, the Freundlich model is more suitable to describing the diesel oil adsorption process, suggesting that the adsorption of biochar to diesel oil is a multi-layer sorption with the heterogeneous adsorption sites on the solid’s surface [55], which may relate to functional groups such as hydroxyl (–OH–) and carboxyl (–COO–) on the surface of CSAB biochar. Applying the Freundlich model, the K_F_ values of CSAB and CSB were 1.0617 and 0.993, respectively. Higher adsorption values indicated that CSAB had better adsorption performance, and CSAB had the greatest adsorption capacity for diesel oil when compared with CSB. The n value was larger than 1, suggesting that diesel oil was favorably adsorbed by crab shell biochar [42]. Moreover, the actual adsorption capacities of the as-prepared samples were higher than the *q_m_* value calculated by the Langmuir equation, which may also suggest the existence of a multi-layered sorption. Several previous studies used the Freundlich isotherm model to describe the oil adsorption behavior of adsorbing materials, such as activated carbon [56], bentonite, deposited carbon [19], and chitosan [42], which were consist with this study.

## 4. Conclusions

An activated biochar (CSAB) derived from crab shell waste was successfully prepared with a great potential application as a promising adsorbent for the removal of diesel oil from oily wastewater. The adsorption experiments showed that the adsorption temperature, contact time, initial pH, and the adsorbent dosage had a great influence on the adsorption performance of diesel oil by crab shell biochar, and optimal experimental conditions were: temperature = 30 °C, adsorbate dosage = 0.2 g, pH = 7, contact time = 4 h. The present investigation on adsorption kinetics demonstrated that CSAB, KOH-modified biochar with mesoporous structures, had a high adsorption capacity of diesel oil (about 93.9 mg/g), which was due to its high specific surface area (2441 m^2^/g), excellent pore volume (1.682 m^3^/g), and unique surface characteristics presented with functional groups such as hydroxyl (–OH–) and carboxyl (–COO–). Results of adsorption isotherm experiments suggested that the actual adsorption process of as-prepared CSAB was interpreted as a multi-layered sorption. Consequently, the low cost and efficient material may have great potential for the uptake of oil pollutants in water.

## Figures and Tables

**Figure 1 materials-12-00236-f001:**
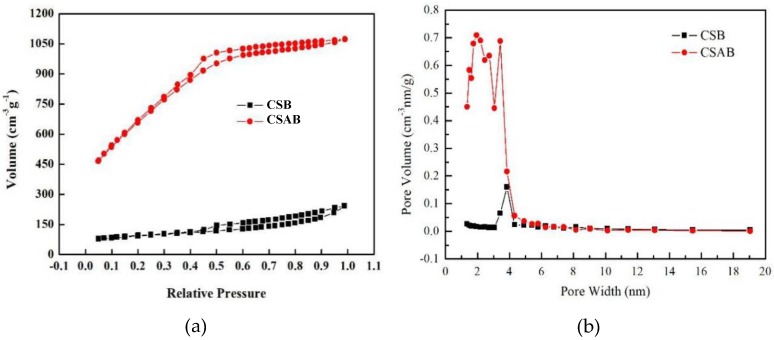
(**a**) Nitrogen adsorption-desorption isotherms; (**b**) pore size distribution.

**Figure 2 materials-12-00236-f002:**
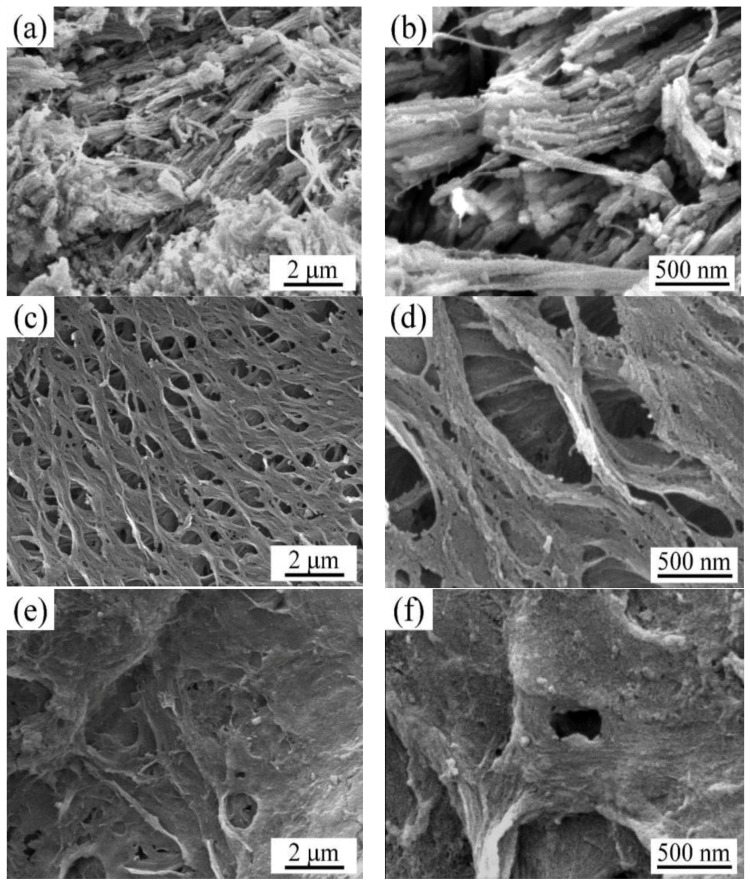
SEM images of (**a**,**b**) crab shell powder; (**c**,**d**) crab shell biochar (CSB); (**e**,**f**) crab shell activated biochar modified by potassium hydroxide (CSAB).

**Figure 3 materials-12-00236-f003:**
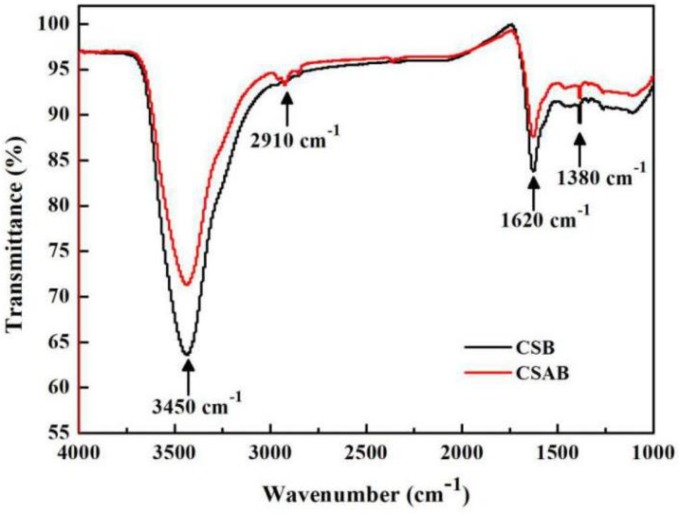
FTIR spectra of CSB and CSAB.

**Figure 4 materials-12-00236-f004:**
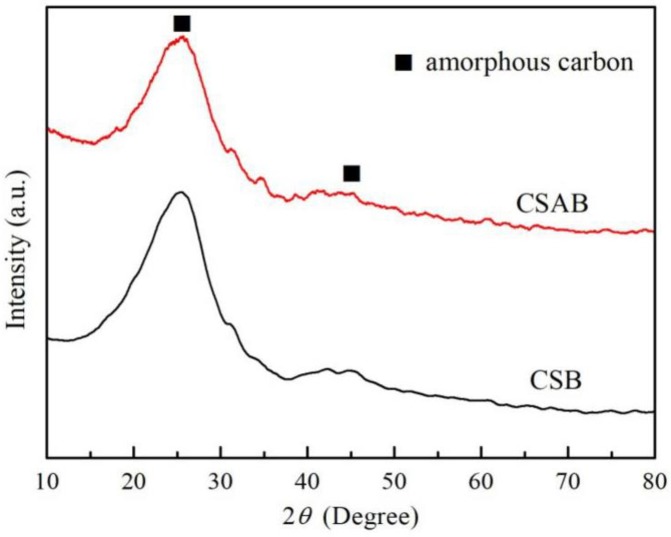
XRD patterns of CSB and CSAB.

**Figure 5 materials-12-00236-f005:**
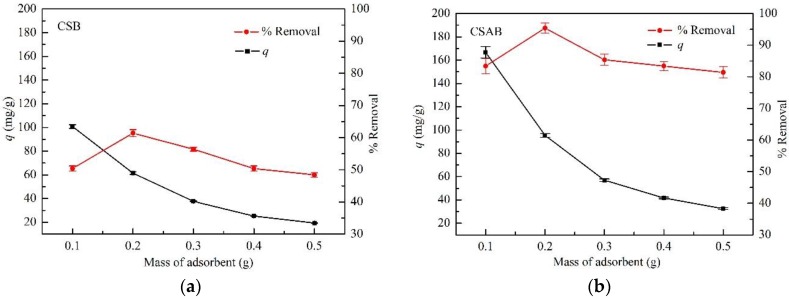
Effect of sorbent dosage on the adsorption of diesel oil onto: (**a**) CSB; (**b**) CSAB.

**Figure 6 materials-12-00236-f006:**
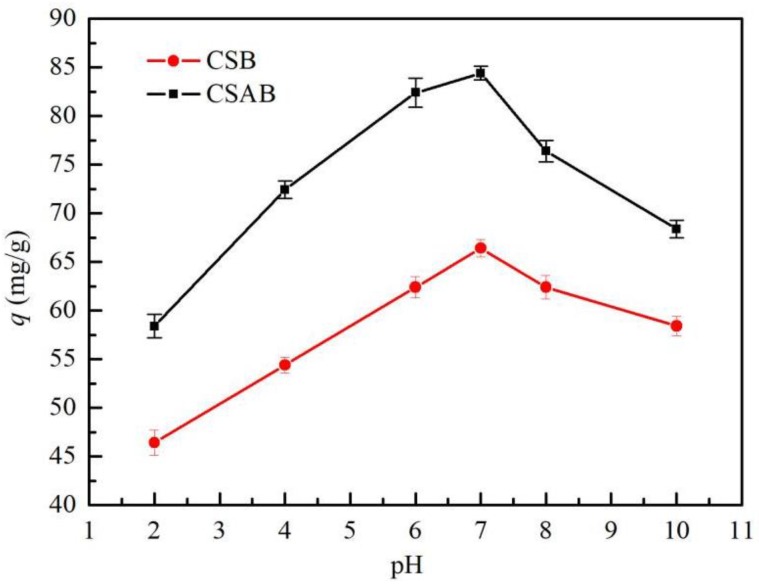
Effect of solution pH on adsorption of diesel oil onto biochar.

**Figure 7 materials-12-00236-f007:**
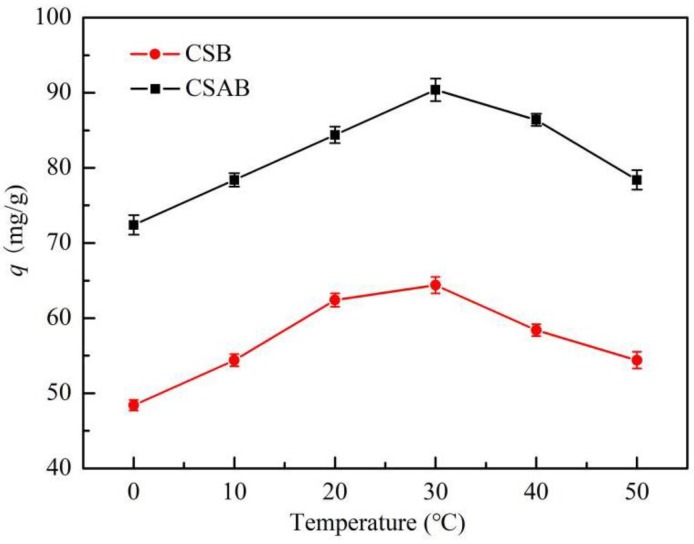
Effect of temperature on adsorption of diesel oil onto biochar.

**Figure 8 materials-12-00236-f008:**
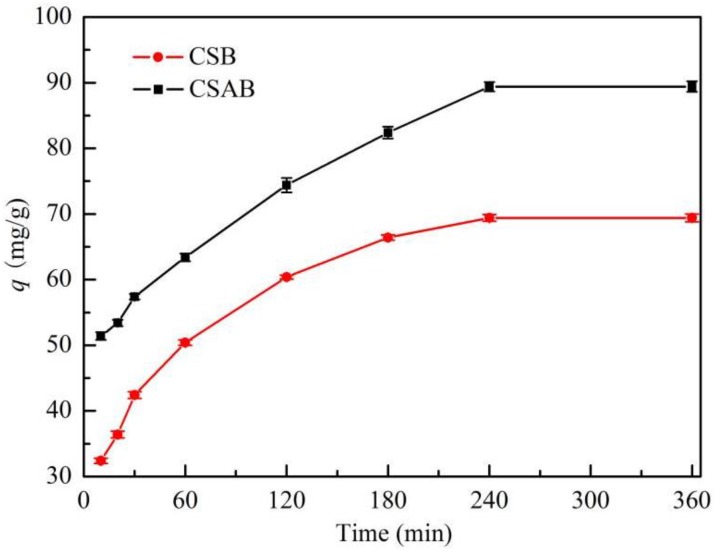
Effect of contact time on the adsorption of diesel oil onto biochar.

**Figure 9 materials-12-00236-f009:**
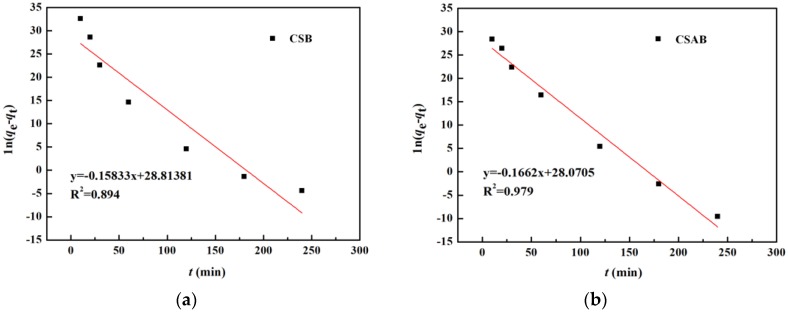
Pseudo first-order model for adsorption of diesel oil onto: (**a**) CSB; (**b**) CSAB.

**Figure 10 materials-12-00236-f010:**
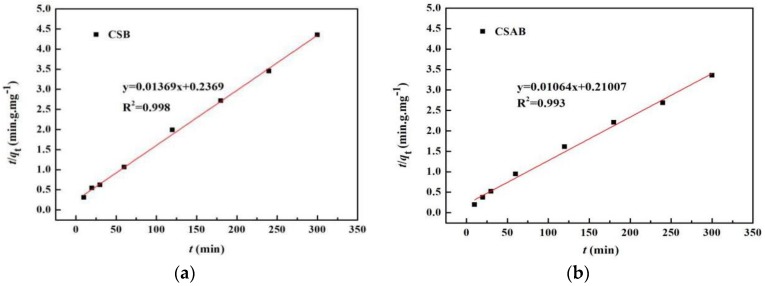
Pseudo second-order model for adsorption of diesel oil onto: (**a**) CSB; (**b**) CSAB.

**Figure 11 materials-12-00236-f011:**
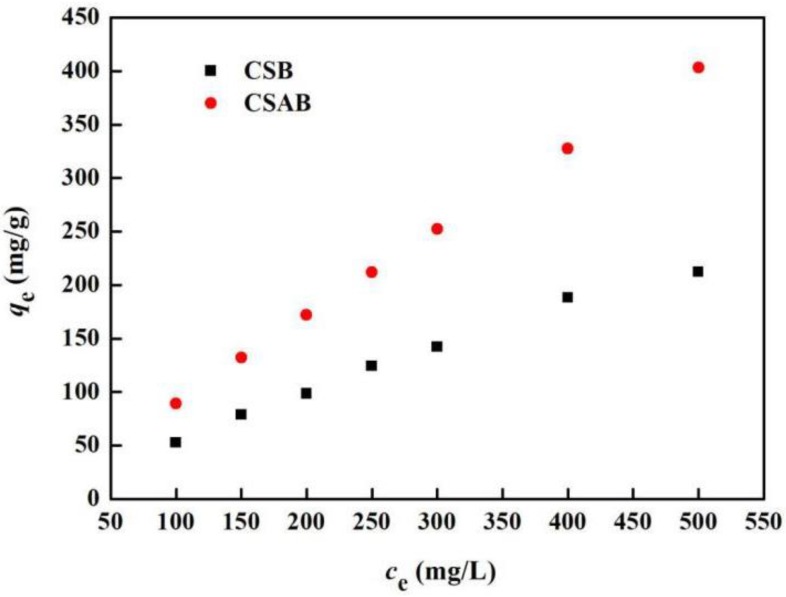
Adsorption isotherms of diesel oil onto crab shell biochars.

**Figure 12 materials-12-00236-f012:**
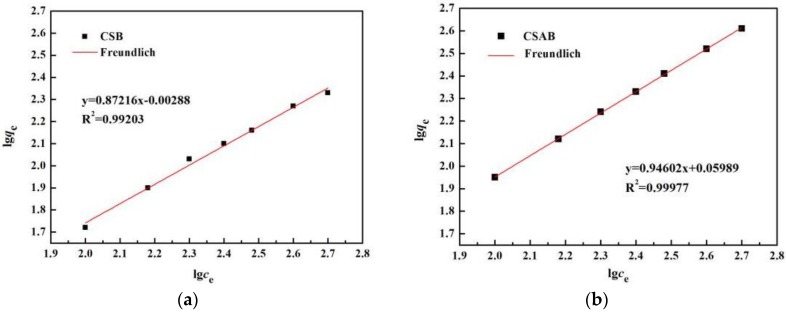
Freundlich isothermal adsorption equation fitting of diesel oil onto: (**a**) CSB, (**b**) CSAB.

**Figure 13 materials-12-00236-f013:**
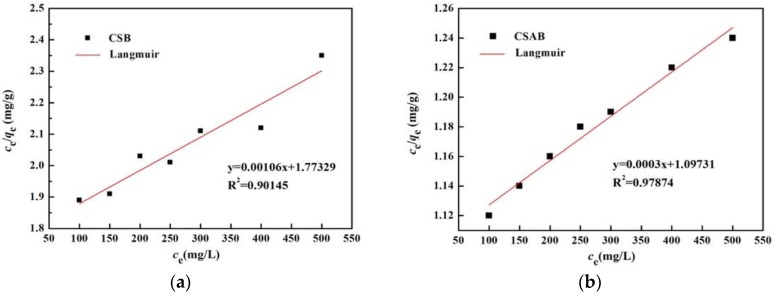
Langmuir isothermal adsorption equation fitting of diesel oil onto: (**a**) CSB, (**b**) CSAB.

**Table 1 materials-12-00236-t001:** BET characteristics of biochar samples.

Sample	Surface Area (m^2^/g)	Pore Volume (m^3^/g)	Pore Size (nm)
CSB	307	0.324	3.840
CSAB	2441	1.682	1.937

**Table 2 materials-12-00236-t002:** Comparison of the pseudo-first order and pseudo-second order models for the adsorption of diesel oil on crab shell biochar.

Sample	Pseudo-First Order			Pseudo-Second Order		
	*q_e_*, 1 (mg/g)	*k*_1_ (min^−1^)	*R* ^2^	*q_e_*, 2 (mg/g)	*k*_2_ (g/(mg·min)	*R* ^2^
CSAB	79.82	0.1662	0.979	93.9	0.023	0.993
CSB	65.02	0.15833	0.894	73.1	0.089	0.998

**Table 3 materials-12-00236-t003:** The thermodynamic parameters in the equation of biochar on diesel crab.

Sample	Freundlich Model			Langmuir Model		
	*n*	*K_F_*	*R* ^2^	*q_m_* (mg/g)	*K_L_*	*R* ^2^
CSAB	1.057	1.0617	0.999	57.74	0.0158	0.978
CSB	1.146	0.993	0.992	30.71	0.0184	0.901

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
