# Peer review of "Effective Adsorption of Diesel Oil by Crab-Shell-Derived Biochar Nanomaterials"

_materials, 2019, doi:10.3390/ma12020236_

Reviewer 1 Report

General comment:

The manuscript deals with investigations on diesel oil removal from oily wastewater by adsorption through crab shell activated biochar modified by KOH impregnation. Crab shell biochar without and with characterization was carried out by SEM, XRD and FTIR. Thermodynamic modelling and kinetic modelling were also carried out.

The manuscript is suitable to be published in this journal; however, some major points should be addressed before publication.

Some minor language mistakes are present that should anyway be corrected.

Please, check section numbering.

1. Introduction

Adsorption is a technique widely used for wastewater treatment for the removal of both organic and inorganic pollutants. I would like to suggest to improve the introduction by including a short overview on wastewater treatment by adsorption. Please, consider the following papers:

o   Erto, A., Chianese, S., Lancia, A., Musmarra, D., 2017. On the mechanism of benzene and toluene adsorption in single-compound and binary systems: Energetic interactions and competitive effects, Desalination and Water Treatment, 86, 259-265.

o   Liu, J., Zhu, X., Zhang, H., Wu, F., Wei, B., Chang, Q., 2018. Superhydrophobic coating on quartz sand filter media for oily wastewaterfiltration, Colloids and Surfaces A: Physicochemical and Engineering Aspects, 553, 509-514.

o   Diaz de Tuesta, J.L., Silva, A.M.T., Faria, J.L., Gomes, H.T., 2018. Removal of Sudan IV from a simulated biphasic oily wastewater by using lipophilic carbon adsorbents, Chemical Engineering Journal, 347, 963-971.

2.4. Adsorption experiments

Please, specify amounts of adsorptive material for the equilibrium investigations and the ones for the kinetic investigations.

Please, detail diesel oil concentration measurement and technology.

Please, specify experimental run times. In particular, please specify the time required for the achievement of the equilibrium condition.

Please, specify if investigations were carried out in duplicate or triplice etc.

The regeneration of adsorptive material should be taken into account and investigated. Some consideration should anyway be included.

3.2 Adsorption kinetics

Please, specify unit of constant kinetics.

Please, check symbol of constant kinetics (the ones used into the text are different that the ones used in equation 1 and equation 2).

Please, use “pseudo-first order” instead of “pseudo-one order”.

Figure 6: please, remove the table.

Please, improve discussion of results and improve comparison between experimental findings and literature data. Moreover, please, compare the performance of the proposed adsorptive material with the performance of other adsorptive materials.

Please specify if the experimental qe values are the ones shown in Figure 7. Otherwise add them.

Please, add standard deviation if available.

3.3 Adsorption isotherms

Please, improve discussion of results and improve comparison between experimental findings and literature data. Moreover, please, compare the performance of the proposed adsorptive material with the performance of other adsorptive materials.

Please, add standard deviation if available.

Author Response

A1: Yes, we have improved the introduction part, and added a brief overview and cited above and some more reference papers.

A2.4: In the part II and III, the effects of adsorbent dosage, adsorption equilibrium time, pH and initial concentration of diesel oil solution on adsorption have been improved, and the reaction conditions of each experiment have been clarified. Please refer to the manuscript.

3.2 Adsorption kinetics

Please, specify unit of constant kinetics.

A: Yes, the unit was fixed, pls refer to the text.

Please, check symbol of constant kinetics (the ones used into the text are different that the ones used in equation 1 and equation 2).

A:Yes the symbol was fixed.

Please, use pseudo-first order instead of pseudo-one order.

A: Yes, it was fixed.

Figure 6: please, remove the table.

A: Yes the table was fixed.

Please, improve discussion of results and improve comparison between experimental findings and literature data. Moreover, please, compare the performance of the proposed adsorptive material with the performance of other adsorptive materials.

Please specify if the experimental qe values are the ones shown in Figure 7. Otherwise add them.

Please, add standard deviation if available.

A: The unit of kinetics was modified, the symbols of the kinetic constants were modified, the writing errors were corrected, and the comparative discussion was added.

Standard deviation analysis was added in the experimental part for investigating the effect of different adsorption conditions on adsorption.

A3.3: Some discussion and relevant comparison have been added, pls refer to the manuscript.

Reviewer 2 Report

Manuscript ID - materials-413610

Effective Adsorption of Diesel Oil by Crab-Shell-Derived Biochar Nanomaterials

General Comments for Authors

I support the idea of finding the new, applicable, and cheap waste-derived bio-sorbent for the removal of contaminants from the wastewaters, since there is a need for an alternative for the commercial costly adsorbents such as activated carbon. This is an advantage of this research. Moreover, the obtained results indicate that the crab shell activated biochar has potential for the removal of oil contaminants from the wastewaters.

But, except the material, there is no such novelty in this work. The characterization of the material, performed adsorption experiment as well as application of the Langmuir and Freundlich isotherms, as well as the pseudo-second-order and the pseudo-first-order kinetic models are very well investigated and covered in the literature. Thus, most of the results could be predicted without experimentation. Activation with KOH is known to improve adsorption properties due to increase in surface area and pore volume.

The advantage will be to find out the process conditions which will be the best and the optimal once (in regard to both efficiency and cost) in the case of application of CSB and CSAB in diesel oil removal from wastewaters. To achieve this and to improve the manuscript, I suggest that Authors upgrade the experiment with more investigation in regard to wider range of pH values, contact times, stirring speeds, mass of the adsorbent (S/L ratio) and temperatures, during adsorption experiment. This will increase the impact of the manuscript and will enable some final conclusions about the cost-effective applicability of crab shell biochar in treatment of oily wastewater. The investigation of only one pH value, S/L ratio, temperature and stirring speed doesn’t give the possibility to find out the optimal process conditions. Since this is a batch study, which is no such time consuming as column process, an examination of the wider range of the experimental conditions does not represent such problem. If Authors already done these investigations, than this should be stated in the manuscript, to justify choice of examined experimental conditions.

Furthermore, I suggest that Authors take into consideration following:

- What about the residual concentration of oil in the aqueous solution? Is it below the maximum permissible level, or the additional degree of purification is required?

- What about the oily adsorbent after the adsorption? What Authors suggested to do with it?

- Did Authors consider the potential for recovery of oil from crab shell biochar through sorption/desorption cycles? It is an important research aspect. Besides the use of low-cost sorbents, their reuse play also important role in increase of cost effectiveness in wastewater treatment.

Detail comments

1. Introduction

Over recent years, large amount of diesel oil hydrocarbon has entered environment as a result of human industrial activities. This sentence needs to be supported by the references. It should be combined with the sentence There are various processes causing oily wastewater, including oil extraction, refining, storage, transportation and production of petrochemical industry [2].

The residue of oil pollutants i natural water has arouse wide attention of researchers [1].Authors should support this sentence with more references. By reading this sentence I have expected several references from recent years or at least one review paper.

2. Materials and Methods

2.2. Preparation of activated carbon

Did Authors try carbonization at different temperatures, and choose this one as optimal one?

2.4. Adsorption experiments

Based on what Authors choose the stirring speed of 150 r/min, pH value of 7, initial concentrations range of oil from 100-500 mg/L and temperature of 30 °C? Did Authors examine in some previous studies other pH values, contact times, stirring speeds and water temperatures during adsorption experiment?

What was the contact time?

What was the S/L ratio? It is no evident what the adsorbent dosage was. This should be stated in the manuscript.

With what Authors adjusted the pH of 7? This should be stated in the manuscript. Why Authors choose pH = 7?

The solution pH as well as the water temperature can significantly affect the sorption capacity.

How to know the optimal experimental conditions without detail investigation of their wide range? Since all these conditions will affect the sorption efficiency.

Namely, the data reported in the literature can be helpful in determining the optimum range for starting the experiment, but not for the certain choice of ideal conditions. They only can be established by detailed experiments.

3. Results

3.1 Structure characterization of crab shell biochar

Figure 1 (b) shows the pore size distribution curve of crab shell biochar obtained by the Barrett-Joyner-Halenda (BJH) method. This method should be supported with the reference.

The pore size distribution (PSDS) of CSB and CSAB also indicates the above results, which are mainly in the range of 0.5 - 20 nm. From the Figure 1(b) the pore width is in the range from 1 nm to 20 nm.

In lines 145-150, when explaining the Figure 2, Authors should highlighted in the text which part of Fig. 2 (a, b, c, d, e or f) they are explaining since in this way is very hard to follow.

3.2 Adsorption kinetics

Authors should support the applied kinetic models with references.

The symbols in the text, tables and in the equations should be equalized here and elsewhere in the manuscript in regard to italic types of letters. Also Authors must defined are the rate constants k1 and k2 or K1 and K2?

In line 186 be careful with the name of the kinetic model, it is pseudo-first not pseudo-one order model.

3.3 Adsorption isotherms

Authors should support the applied adsorption isotherms with references. Again, is it kL or KL?

In Eq. (4) the lg should be corrected in log.

What about the Figure 7? When the adsorption achieved equilibrium?

Can the adsorption capacity for CSAB of 403.3 mg/g be called as the maximum one? It is the highest one for examined experimental conditions, but not the maximum one, since the equilibrium wasn’t reached in the case of CSAB for this range of initial oil concentrations.

Where is the section 4. Discussion?

After 3. Results comes 5. Conclusions.

It means that section 3. are the Results and Discussion, together in one section. Authors should correct this.

Author Response

In general, many thanks for your kind suggestion! This is our preliminary research work, we will carry out further research into this issue systematically in the future. We will continue to publish some research results.

A0: The diesel oil removal performance of crab shell biochar nanomaterials is very stable. However, the amount of as-prepared samples was small due to the limitation of experimental conditions at that time. The recovery of sample powders was so little while the loss was so great. In addition, repeated experiments would take a long time.

In the future research, we will plan to prepare samples into other forms or to find a support material which is easy to recover and to handle with for a wider application, such as immobilization, film-like structure, addition of magnetic particles, etc.

Detail comments:

A1: Yes, we have combined these two sentences and provide more review papers accordingly.

A2.2: Yes, on the optimum carbonization temperature of crab shell we have done a lot of experiments in the past. The experimental results show that high specific surface area and fine microstructure of biochar can be obtained by calcination at 700°C.

A2.4: Yes, in order to answer your questions, we have added initial pH, temperature, sorbent dosage, and contact time optimization experiments in the manuscript, pls refer to the text.

A3.1: Thanks, Reference of BJH method has been added; Aperture distribution is 1-20 nm, 0.5 is a typing error; The corresponding description of Figure 2 has been marked.

A3.2: Yes, we have added reference in the text accordingly, we choose k1 and k2 and other modifications have been made.

A3.3: Yes, we have revised the text accordingly to answer your questions. We have added discussions in the manuscript, and changed “5” to “4” in the text.

Reviewer 3 Report

The paper should be improved by studying results of adsorbed samples.

The adsorbents characterization is OK but the interaction in the adsorbed samples

can be improved by FT-IR study, DSC and DRX.

They have to study the desorption too.

Author Response

The purpose of this study is to find an inexpensive oil-absorbing material to treat oily wastewater. After a lot of research, we have found that abandoned crab shell can be utilised for an effective oil-absorbing material. Crab shell biochar with super high specific surface area has been prepared by anaerobic calcination and KOH activation. Its structure and morphology were characterized by BET, FTIR, SEM and XRD first time in our laboratory. The adsorption process has been studied theoretically by adsorption isotherm and adsorption kinetics. The adsorption was mainly achieved by chemical adsorption and multi-layer adsorption. The maximum adsorption capacity of diesel oil so far is 93.9 mg/g, which has proved that crab shell biomass can be a promising, cheap and environmental friendly oil-absorbing material.

Studying results of adsorbed samples is a very important scientific issue. We will carry out further research into this issue systematically in the future. We will continue to publish some research results. Thanks again for your kind suggestion.

Regarding the desorption experiment: The diesel oil removal performance of crab shell biochar nanomaterials is very stable in our experiments. However, the amount of samples prepared is very small due to the limitation of experimental conditions. The recovery of sample powders is so little and it is not so easy while the loss is very large. In addition, repeated experiments will take a long time to carry out. In the future research, we will plan to prepare samples into other forms or to find a support material which is easy to recover and to handle with for a wider application, such as immobilization, film-like structure, addition of magnetic particles, etc.

Round  2

Reviewer 1 Report

The authors modified the manuscript according to my suggestion. The paper can be published.

Author Response

OK,thanks!

Reviewer 2 Report

Manuscript ID materials-413610

Effective Adsorption of Diesel Oil by Crab-Shell-Derived Biochar Nanomaterials

Comments for Authors

The manuscript is greatly improved especially in regard to more experimental results. Still, the spell check is required, and some errors in the text should be corrected, so please pay attention to the following:

2. Materials and Methods

2.4. Adsorption experiments

Line 114: Adsorption equilibrium experiments were conducted in bath mode in a series of 250 ml…instead of bath mode did Authors mean batch mode?

Lines 124-125: The initial pH of the reaction solution was adjusted from 5 to 9 by adding either diluted HCl or NaOH (0.1 mol/L)…the range of the initial pH values was from 2.0 to 10.0, according to Figure 5 (the exact pH values were 2.0; 4.0; 6.0; 7.0; 8.0 and 10.0). Authors must correct this in the text.

There is no range of contact time in Materials and Methods section. Is it from 10 to 360 min (as can be seen in Figure 7)? It should be added here in accordance to all other ranges of investigated experimental conditions.

3. Results

Regarding the results, I have suggestion: since the Authors chooses the adsorbent dosage of 0.2 g in investigation of different temperatures, I suppose due to the highest adsorption efficiency in Figure 8, it seems more appropriate to present first the results of different adsorbent dosage and afterwards the effect of temperature.

Also, in titles of Figures 5, 6, 7 and 8, as well as in the text which explains the results in these figures, it should be great to add the other experimental conditions which were kept constant. It will be much easier to follow.

3.3 Effect of adsorbent temperature

It would be good somewhere in the text to add the S/L ratio.

3.5 Effect of sorbent dosage

The Authors should correct the sorbent dosage in the title into the adsorbent dosage.

Also, it should be better to express the adsorbent dosage in the text in grams as it is in Figure 8, as well as in the Materials and Methods section.

Figure 8 - Please check the legends in Figure 8a and b. Are they correct? Moreover, since the Figures 8a and 8b are one by one, it would be much easier to compare them if the partition on y axis is equal. I assume that in Figure 8a the max. value on y axis of 200 mg/g want make the results invisible, and it will be comparable with Figure 8b, Moreover, the differences between CSB and CSAB will be noticeable immediately.

Again, since in the Results section the Authors also discuss, this should be the section 3. Results and Discussion.

4. Conclusions

Authors should add in this section the optimal experimental conditions (based on performed investigations) of the range of different pH, initial concentration, adsorbent dosage and contact time.

Author Response

2. Materials and Methods

2.4. Adsorption experiments 

Line 114: Adsorption equilibrium experiments were conducted in bath mode in a series of 250 ml…instead of bath mode did Authors mean batch mode?

A: Thanks. We are sorry for the mistakes. We have revised the whole manuscript carefully and tried to avoid any grammar or syntax error. The main modifications and revisions have been highlighted by red color in the revised manuscript.

Lines 124-125: The initial pH of the reaction solution was adjusted from 5 to 9 by adding either diluted HCl or NaOH (0.1 mol/L)…the range of the initial pH values was from 2.0 to 10.0, according to Figure 5 (the exact pH values were 2.0; 4.0; 6.0; 7.0; 8.0 and 10.0). Authors must correct this in the text.

A: We appreciate your advice. We have revised it in the manuscript which was highlighted by red color.

There is no range of contact time in Materials and Methods section. Is it from 10 to 360 min (as can be seen in Figure 7)? It should be added here in accordance to all other ranges of investigated experimental conditions.

A: Yes, we have added the range of contact time in materials and methods section.

3. Results 

Regarding the results, I have suggestion: since the Authors chooses the adsorbent dosage of 0.2 g in investigation of different temperatures, I suppose due to the highest adsorption efficiency in Figure 8, it seems more appropriate to present first the results of different adsorbent dosage and afterwards the effect of temperature.

A: Yes, experiments on the effect of dosage on adsorption have been put at the first place.

Also, in titles of Figures 5, 6, 7 and 8, as well as in the text which explains the results in these figures, it should be great to add the other experimental conditions which were kept constant. It will be much easier to follow.

A: Yes, the experimental conditions were addressed clearly in each section.

3.3 Effect of adsorbent temperature 

It would be good somewhere in the text to add the S/L ratio.

A: Yes, the S/L ratio has been added in section 3.2.

3.5 Effect of sorbent dosage 

The Authors should correct the sorbent dosage in the title into the adsorbent dosage.

A: we have revised in the manuscript according to your suggestion.

Also, it should be better to express the adsorbent dosage in the text in grams as it is in Figure 8, as well as in the Materials and Methods section.

AThanks for your suggestion, we have expressed the adsorbent dosage in the text in grams.

Figure 8 - Please check the legends in Figure 8a and b. Are they correct? Moreover, since the Figures 8a and 8b are one by one, it would be much easier to compare them if the partition on y axis is equal. I assume that in Figure 8a the max. value on y axis of 200 mg/g want make the results invisible, and it will be comparable with Figure 8b, Moreover, the differences between CSB and CSAB will be noticeable immediately.

 A: Thanks for your suggestion. We have revised Figure 8, pls check it in the manuscript.

Again, since in the Results section the Authors also discuss, this should be the section 3. Results and Discussion.

A: We have added “discussion” for the section 3.

 4. Conclusions

Authors should add in this section the optimal experimental conditions (based on performed investigations) of the range of different pH, initial concentration, adsorbent dosage and contact time.

A: The adsorption experiment showed that the adsorption temperature, contact time, initial pH and the adsorbent dosage had a great influence on the adsorption performance of diesel oil by crab shell biochar, optimal experimental conditions were: temperature = 30 °C,adsorbate dosage = 0.2g,pH = 7,contact time = 4 h.